



# Simulating low-frequency wind fluctuations

Abdul Haseeb Syed[1] and Jakob Mann[1]

[1]Department of Wind and Energy Systems, Technical University of Denmark, 4000 Roskilde, Denmark

**Correspondence:** Abdul Haseeb Syed (absy@dtu.dk)

**Abstract.** Large-scale flow structures are vital in influencing the dynamic response of floating wind turbines and wake meandering behind large offshore wind turbines. It is imperative that we develop an inflow wind turbulence model capable of replicating the large-scale and low-frequency wind fluctuations occurring in the marine atmosphere since the current turbulence models do not account well for this phenomenon. Here, we present a method to simulate low-frequency wind fluctuations. This method employs the two-dimensional spectral tensor for low-frequency, anisotropic wind fluctuations presented by Syed and Mann (2023) to generate stochastic wind fields. The simulation method generates large-scale 2D wind fields for the longitudinal $u$ and lateral $v$ wind components. The low-frequency wind turbulence is assumed to be independent of the high-frequency turbulence thus a broad spectral representation can be obtained just by superposing the two turbulent wind fields. The method is tested by comparing the simulated and theoretical spectra and co-coherences of the combined low- and high-frequency fluctuations. Furthermore, the low-frequency wind fluctuations can also be subjected to anisotropy. The resulting wind fields from this method can be used to analyze the impact of low-frequency wind fluctuations on wind turbine loads and dynamic response and for studying the wake meandering behind large offshore wind farms.

## 1 Introduction

Several models are available for generating high-frequency wind fluctuations within a three-dimensional space. These models can generate realistic wind fields that can be used for load estimation on structures such as bridges, wind turbines, and buildings. For wind turbine design and load calculations, the International Electrotechnical Commission (IEC) standards (IEC, 2019) recommend two commonly used models: the Mann uniform shear model (Mann, 1994, 1998) and Kaimal spectral and coherence model (Kaimal et al., 1972; Veers, 1988). A notable advantage of these two models is simulating realistic small-scale turbulence without exorbitant computational time and resources. In contrast, Large Eddy Simulation (LES) or other numerical solutions of the Navier-Stokes equations have proven to be computationally expensive and unfeasible for the wind turbine design process.

While high-frequency fluctuations have more influence on the stresses and fatigue loads experienced by the blades and tower of a wind turbine, low-frequency fluctuations can have a significant effect on the overall energy production and capacity factor of a wind farm. In the context of floating offshore wind turbines, low-frequency wind fluctuations may be of significant importance in terms of dynamic response and loading since these structures can have very low natural frequencies (Nybø et al., 2022). Low-frequency fluctuations are also crucial for the meandering of wakes behind wind farms, which affect power



fluctuations and dynamic loads. The dynamic wake meandering model of Larsen et al. (2008) uses the low-frequency turbulence to move the wake deficit, but it uses a normal turbulence spectrum that does not take into account the excess power spectral energy at low frequencies often seen offshore (Sathe et al., 2013; De Maré and Mann, 2014; Cheynet et al., 2018). Thus, we need a fast method for simulating low-frequency wind fluctuations that are realistic and can be easily integrated with high-frequency wind fields to get a comprehensive spectral range representation.

Here, we present a method for simulating low-frequency wind fluctuations based on the two-dimensional spectral tensor introduced in Syed and Mann (2023). At low frequencies, only the longitudinal ($u$) and lateral ($v$) wind components have strong fluctuations since, at least close to the ground, the presence of the land or sea blocks the vertical large-scale movements. Thus, the vertical wind ($w$) fluctuations at low frequencies attenuate or weaken considerably, rendering the turbulence two-dimensional (2D). The 2D turbulence model only describes the $u$ and $v$ fluctuations in the low-frequency range and assumes that these fluctuations do not vary in the vertical direction. The algorithm to generate stochastic wind fields from the 2D turbulence model is similar to the one described in Mann (1998). The 2D wind field is represented as a discrete Fourier series, which takes the mean squared amplitude of the Fourier coefficients from the 2D spectral model. These coefficients are then multiplied by a random Gaussian field. Subsequently, the resulting product's inverse discrete Fourier transform yields the stochastic wind field.

Section 2 of this paper describes the low-frequency, 2D turbulence model, along with model validation details. Section 3 outlines the process for simulating 2D wind fields containing 2D turbulence. Section 4 describes the process of combining 2D and 3D wind fluctuations to create turbulence boxes that represent a wide spectral range. Finally, a discussion regarding the effect of anisotropy on the 2D turbulence and some basic guidelines to generate 2D wind fields for the wind turbine design load process is presented in Section 5.

## 2 Low-frequency turbulence model

The two-dimensional, incompressible, and isotropic turbulence has the spectral tensor form of (Batchelor, 1953)

$$\phi_{ij}(k_1, k_2) = \frac{E(k)}{\pi k} \left( \delta_{ij} - \frac{k_i k_j}{k^2} \right) \ , \tag{1}$$

where $E(k)$ is the energy spectrum, $k$ is the magnitude of the horizontal wave vector $k = |\boldsymbol{k}| = \sqrt{k_1^2 + k_2^2}$ and $\delta_{ij}$ is the Kronecker delta. The assumption of incompressibility is an approximation. Alcayaga et al. (2022) observe some divergence in a horizontal plane at wind turbine relevant heights. We assume that the energy spectrum is given by

$$E(k) = \frac{ck^3}{\left( L_{2D}^{-2} + k^2 \right)^{7/3}} \ , \tag{2}$$

where $c$ is a constant and a scaling parameter, and $L_{2D}$ is the corresponding length scale of low-frequency fluctuations. This particular shape of (2) is inspired by the von Kármán (1948) spectra. The variance of any horizontal velocity component can



be found by

$$\sigma^2 = \sigma_{11}^2 = \sigma_{22}^2 = \int_0^\infty E(k) = \frac{9}{8} c L_{2D}^{2/3} \ . \tag{3}$$

Due to isotropy, the variance is the same for both wind components. Now, let us introduce scale-independent anisotropy in the energy spectrum. We replace the horizontal wave vector $k = |\boldsymbol{k}| = \sqrt{k_1^2 + k_2^2}$ with $\kappa$ where

$$\kappa^2 = 2(k_1^2 \cos^2 \psi + k_2^2 \sin^2 \psi) \ , \tag{4}$$

and $0 < \psi < \pi/2$ is the anisotropy parameter. Now, the energy spectrum with anisotropy parameter takes the form of

$$E(\kappa) = \frac{c\kappa^3}{\left(L_{2D}^{-2} + \kappa^2\right)^{7/3}} \ . \tag{5}$$

When $\psi = \pi/4$, $k = \kappa$ and (5) takes the form in (2). By inserting $E(\kappa)$ into (1) we can obtain two-point cross spectra $\chi_{ij}^{2D}$ and one-point spectra $F_{ij}^{2D}$ using

$$\chi_{ij}^{2D}(k_1, \Delta y) = \int_{-\infty}^\infty \phi_{ij}(k_1, k_2) \exp(-ik_2 \Delta y) dk_2 \ , \tag{6}$$

where $F_{ij}^{2D}(k_1) = \chi_{ij}^{2D}(k_1, 0)$ is the one-point cross- or auto-spectrum depending on whether the component indices $i$ and $j$ are different or equal. The anisotropy parameter $\psi$ determines the spectral distortion in the wavenumber domain and the spectrum magnitudes of longitudinal and transverse wind components. When the 2D turbulence is isotropic ($\psi = \pi/4$), $F_{11}^{2D} = \frac{3}{5} F_{22}^{2D}$ in the $k_1^{-5/3}$ range. For the anisotropic cases, the ratio can be found using

$$\frac{F_{11}^{2D}}{F_{22}^{2D}} = \frac{3}{5} \cot^2 \psi \ . \tag{7}$$

The energy spectrum must be attenuated at the wavenumbers corresponding to small-scale 3D turbulence. This is necessary because we assume that low-frequency fluctuations are independent of high-frequency fluctuations, and at very high wavenumbers only small-scale 3D turbulence is present. This high wavenumbers range is referred to as the inertial subrange. The turbulence is isotropic in this range and follows a power law (Pope, 2000). For practical reasons, we attenuate the low-frequency turbulence at wavenumbers higher than $1/z_i$ where $z_i$ is the boundary-layer height. This implies that any eddy having a length scale smaller than the boundary-layer height would solely be considered as 3D turbulence. The attenuated $E(\kappa)$ can be defined as

$$E(\kappa) = \frac{c\kappa^3}{\left(L_{2D}^{-2} + \kappa^2\right)^{7/3}} \frac{1}{1 + \kappa^2 z_i^2} \ . \tag{8}$$

Here, the attenuation factor $1/(1 + \kappa^2 z_i^2)$ is an activation function that ensures the energy spectrum smoothly drops to zero for wavenumbers greater than $1/z_i$. This drop is accelerated due to an increased negative slope of the spectrum for $\kappa > 1/z_i$, i.e., $E(\kappa) \propto \kappa^{-11/3}$. Sigmoid functions such as a hyperbolic tangent or a logistic function can also be used as an attenuation factor.





From (6) we can obtain $F_{ij}^{2D}$ as following

$$F_{11}^{2D}(k_1) = c\left(\frac{\Gamma\left(\frac{11}{6}\right) L_{2D}^{\frac{11}{3}}\left\{-p\,_2F_1\left(\frac{5}{6},1;\frac{1}{6};p\right) - 7\,_2F_1\left(\frac{5}{6},1;\frac{1}{2};p\right) + 2\,_2F_1\left(-\frac{1}{6},1;\frac{1}{2};p\right)\right\}}{10\sqrt{2\pi}\,\Gamma\left(\frac{7}{3}\right)(L_{2D}^2 - z_i^2)\sin^3(\psi)\,a^{\frac{5}{6}}}\right.$$
$$\left. + \frac{L_{2D}^{\frac{14}{3}}\sqrt{b}}{2\sqrt{2}\,d^{\frac{7}{3}}\,z_i^3\,\sin^3(\psi)}\right), \tag{9}$$

and

$$F_{22}^{2D}(k_1) = ck_1^2\left(-\frac{z_i^4\,a^{\frac{1}{6}}\,L_{2D}^{\frac{11}{3}}\,\Gamma\left(\frac{17}{6}\right)}{55\sqrt{2\pi}\,(L_{2D}^2 - z_i^2)^2\,b\,\Gamma\left(\frac{7}{3}\right)\sin(\psi)}\left(-9 - 26\,_2F_1\left(-\frac{1}{6},1;\frac{1}{2};p\right)\right.\right.$$
$$+ p^2\left\{15 - 30\,_2F_1\left(-\frac{1}{6},1;\frac{1}{2};p\right) - 59\,_2F_1\left(\frac{5}{6},1;\frac{1}{2};p\right)\right\}$$
$$+ 35\,_2F_1\left(\frac{5}{6},1;\frac{1}{2};p\right) + 15p^3\,_2F_1\left(\frac{5}{6},1;\frac{1}{2};p\right)$$
$$\left.+ p\left\{-54 + 88\,_2F_1\left(-\frac{1}{6},1;\frac{1}{2};p\right) + 9\,_2F_1\left(\frac{5}{6},1;\frac{1}{2};p\right)\right\}\right)$$
$$\left.- \frac{L_{2D}^{\frac{14}{3}}}{\sqrt{2b}\,d^{\frac{7}{3}}\,z_i\,\sin(\psi)}\right), \tag{10}$$

where

$$a = 1 + 2k_1^2 L_{2D}^2 \cos^2(\psi),$$
$$b = 1 + 2k_1^2 z_i^2 \cos^2(\psi),$$
$$p = \frac{L_{2D}^2\,b}{z_i^2\,a},$$

$\Gamma$ is the Gamma function and $_2F_1$ is the hypergeometric function. The two-point cross spectra $\chi_{11}^{2D}(k_1, \Delta y)$ and $\chi_{22}^{2D}(k_1, \Delta y)$ for the attenuated energy spectrum in (8) to our knowledge do not have any analytical solution but can be obtained through numerical integration techniques. An example of $F_{11}$ with and without attenuation at high wavenumbers is shown in Fig. 1.

The 2D turbulence model (Syed and Mann, 2023) combined with the Mann Uniform Shear model for 3D turbulence was validated against measurements from two offshore sites: 10 Hz ultrasonic measurements from FINO1 research platform in the North Sea and line-of-sight (LOS) wind measurements from a forward-looking nacelle lidar in the Hywind Scotland offshore wind farm. A good agreement was recorded between observed and predicted auto-spectra, cross-spectra, and co-coherences. The measured data was classified into different atmospheric stability classes, and it was found that for a 1-hr time series, the low-frequency fluctuations existed in all stability classes. Albeit, the relative strength of 2D turbulence, compared to 3D turbulence, was more dominant during stable stratification. For the 1-hr time series, the mesoscale turbulence peak corresponding to $L_{2D}$ was also not observed. The low-frequency turbulence was in the $F(k) \propto k_1^{-5/3}$ range at both sites. For the FINO1 site, the measured value of $\psi$ was close to $45°$ in the low-frequency range, representing isotropic 2D turbulence. At Hywind Scotland, we observed $\psi < 40°$ reflecting the anisotropy in the 2D turbulence.

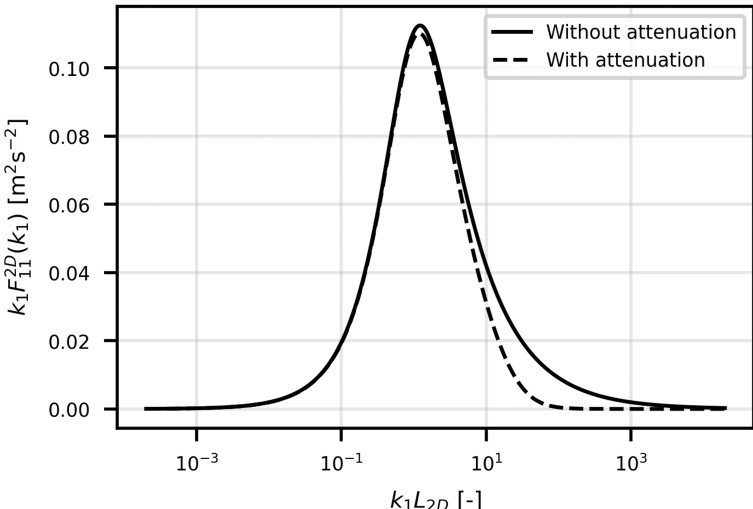

**Figure 1.** Effect of attenuation at high wavenumbers on $F_{11}$ spectrum. Here the model parameters are: $L_{2D} = 20$ km, $z_i = 500$ m, and $\psi = 43°$

In summary, the low-frequency turbulence model has four input parameters:

1. $\sigma_{2D}^2$ the variance exhibited by low-frequency fluctuations (excluding the attenuation),

2. $L_{2D}$ the length scale corresponding to the peak of mesoscale turbulence,

3. $\psi$ the anisotropy parameter, and

4. $z_i$ the attenuation length, which is assumed to be the boundary layer height.

## 3   2D wind field simulation

Here, we will follow the recipe of Mann (1998) to simulate low-frequency, anisotropic wind fields. The 2D turbulence is assumed to be statistically homogeneous in horizontal directions and constant in the vertical direction. Taylor's frozen turbulence
hypothesis is also employed to convert the frequency domain into the wavenumber domain. The wind field will be simulated on a horizontal grid with $N_1$ and $N_2$ grid points in the longitudinal and transverse directions, respectively. The length of the grid in two directions would be $L_1 = N_1 \cdot dx$ and $L_2 = N_2 \cdot dy$. Following Mann (1998), the incompressible, homogeneous, two-dimensional velocity field can be written as a sum of discrete Fourier modes:

$$u_i(\boldsymbol{x}) = \sum_{\boldsymbol{k}} \exp(\iota \boldsymbol{k} \cdot \boldsymbol{x}) C_{ij}(\boldsymbol{k}) n_j(\boldsymbol{k}) \,, \tag{11}$$





where $\sum_{\boldsymbol{k}}$ denotes the sum over all wavevectors $\boldsymbol{k}$, where $k_i = m2\pi/L_i$ for $m = -N_i/2, ..., N_i/2$. $C_{ij}(\boldsymbol{k})$ are the Fourier coefficients, and $n_j(\boldsymbol{k})$ are independent Gaussian stochastic variables. The solution to (11) is approximately

$$C_{ij}(\boldsymbol{k})n_j(\boldsymbol{k}) = \frac{1}{L_1 L_2} \int_A u_i(\boldsymbol{x}) \exp(-\iota \boldsymbol{k} \cdot \boldsymbol{x}) d\boldsymbol{x} , \tag{12}$$

where $\int_A d\boldsymbol{x}$ is integration over the area $L_1 \times L_2$. The process of obtaining $C_{ij}$ involves multiplying (12) with its complex conjugate which gives

$$C_{ij}^*(\boldsymbol{k})C_{ij}(\boldsymbol{k}) = \int \phi_{ij}(\boldsymbol{k}') \prod_{(l=1)}^{2} \operatorname{sinc}^2\left(\frac{(k_l - k_l')L_l}{2}\right) d\boldsymbol{k}' , \tag{13}$$

where $\operatorname{sinc} x \equiv (\sin x)/x$. In the case if $L_l \gg L_{2D}$ where $l = 1, 2$ , (13) can be simplified to

$$C_{ij}^*(\boldsymbol{k})C_{ij}(\boldsymbol{k}) = \frac{(2\pi)^2}{L_1 L_2} \phi_{ij}(\boldsymbol{k}) \tag{14}$$

The length scale $L_{2D}$ corresponding to the mesoscale turbulence peak is quite large, usually in the order of $10^5$ to $10^6$ m. Simulating a high-resolution wind field containing the wavenumbers corresponding to $L_{2D}$ would be costly in terms of computation time. Usually, $L_2 \ll L_{2D}$ when simulating wind fields for single wind turbine load calculations. So, the simplified relation in (14) no longer holds true. We have observed that if we simplify the $\operatorname{sinc}^2$ function for $L_1$ and replace it with $2\pi/L_1$ but integrate the $\operatorname{sinc}^2$ function for $L_2$, we would get simulated spectra much closer to the target spectra.

$$C_{ij}^*(\boldsymbol{k})C_{ij}(\boldsymbol{k}) = \frac{2\pi}{L_1} \int \phi_{ij}(k_1, k_2') \operatorname{sinc}^2\left(\frac{(k_2 - k_2')L_2}{2}\right) dk_2' . \tag{15}$$

To speed up the numerical integration, the limits of integration are $k_2 - 2\pi/L_2$ to $k_2 + 2\pi/L_2$. A correction factor is multiplied, compensating the loss in variance due to the limited integration interval. This problem with discretization has been discussed in detail by Mann (1998). The Fourier coefficients obtained from (14) or (15) after taking a matrix square root are then multiplied by a random Gaussian field. The resulting product's inverse discrete Fourier transform would yield the wind field. Figure 2 illustrates the effect of discretization on the simulated spectra. In Fig. 2(b) when $L_2 \ll L_{2D}$, $C_{ij}(\boldsymbol{k})$ obtained via (14) underestimate $F_{11}(k_1)$ and overestimate $F_{22}(k_1)$ at very low $k_1$ values. In such cases, $C_{ij}(\boldsymbol{k})$ must be evaluated using (15).

Figure 3 shows the simulated $u$ and $v$ low-frequency fluctuations, where the input parameters are: $L_{2D} = 15$ km, $\sigma^2 = 0.6$ m$^2$s$^{-2}$, $z_i = 500$ m, and $\psi = 43°$. These parameters, with the exception of $L_{2D}$, are representative of typical neutral conditions for $8 < \overline{U} < 10$ ms$^{-1}$ at the FINO1 offshore site. Here, large-scale coherent structures can be identified along the longitudinal axis for the $u$ component. We can also observe the almost equal variance in the $u$ and $v$ fluctuations due to $\psi$ being close to the isotropic value of $45°$. The one-dimensional spectra of this simulated 2D wind field are illustrated in Fig. 6 (a). The spectra derived from the simulated wind field are in excellent agreement with the theoretical spectra mentioned in (9) and (10). Normalized cross spectra (co-coherence, the real part of the cross-spectrum divided by the auto-spectrum) for the simulated 2D wind field components are also compared with the theoretical expression in 6. In Fig. 4, co-coherence of $u$ and $v$ is plotted as a function of $k_1$ for separations ranging from 750 m to 7500 m. Once the lateral separation distance $\Delta y$ approaches $L_{2D}$ (in this case 15 km), the normalized cross-spectra decreases significantly.

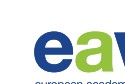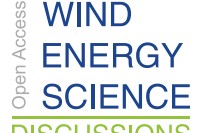

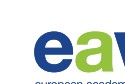

**Figure 2.** Simulated and target $F_{11}$ and $F_{22}$ spectra for 2D rectangular grids having dimensions of (a) $40L_{2D} \times 5L_{2D}$ and (b) $L_{2D} \times 0.125L_{2D}$. Solid lines represent the target spectrum, dashed lines represent simulated spectra from $C_{ij}(\boldsymbol{k})$ obtained using (14), and dash-dot lines represent $C_{ij}(\boldsymbol{k})$ obtained using (15). The simulated spectra are obtained from the mean of 10 realizations. Other parameters are: $\sigma^2 = 2 \, \mathrm{m^2 s^{-2}}$, $z_i = 500$ m, and $\psi = 45°$





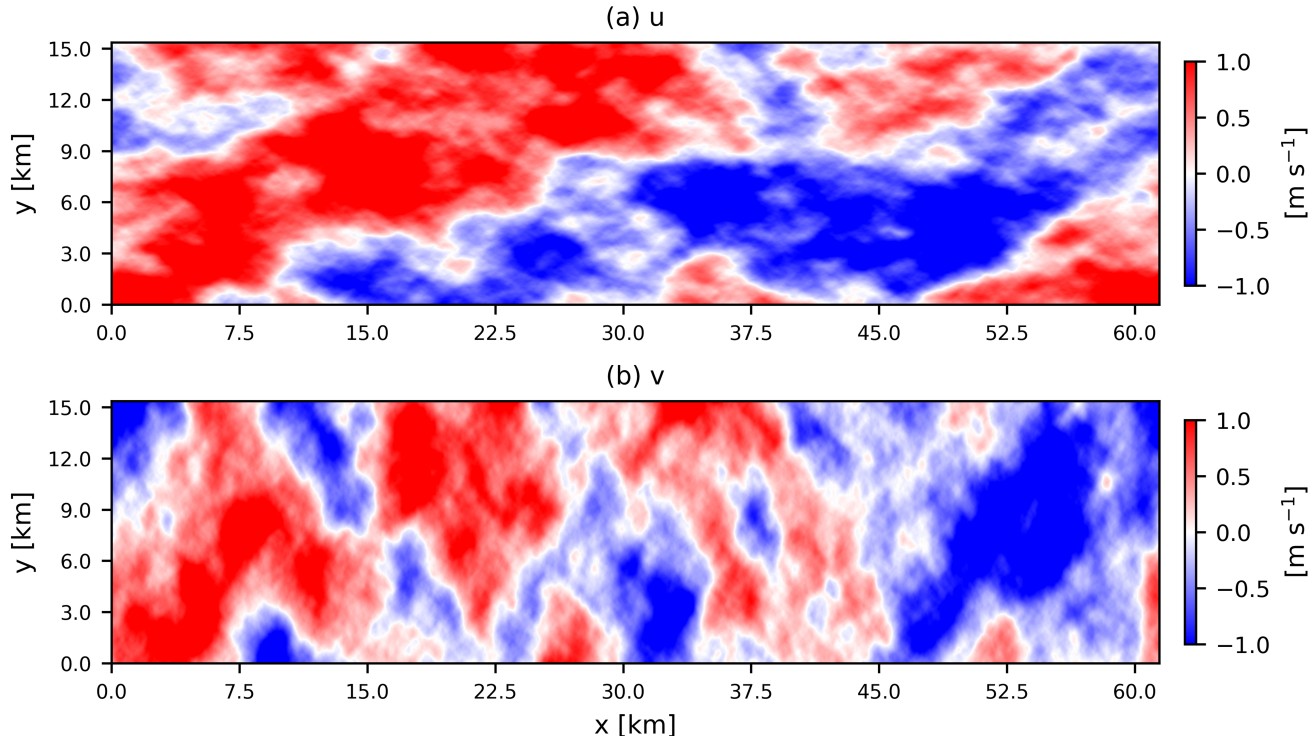

**Figure 3.** Simulated low-frequency fluctuations of longitudinal $u$ and transverse $v$ wind components. Here the input parameters are: $L_{2D} = 15$ km, $\sigma^2 = 0.6$ m$^2$s$^{-2}$, $z_i = 500$ m, and $\psi = 43°$

## 4 Combining 2D and 3D fluctuations

Mann (1994) presented the Uniform shear model for small-scale turbulence in the neutral atmosphere. We will combine the two models assuming that the large-scale and small-scale fluctuations are independent. Figure 5 displays a combined 2D and 3D turbulence wind field of $u$ and $v$ wind components for relatively smaller dimensions. The 3D wind field is generated by the Mann uniform shear turbulence model, which has three input parameters: the dissipation parameter $\alpha\epsilon^{2/3} = 0.01$ m$^{4/3}$s$^{-2}$, the turbulence length scale $L_{3D} = 50$ m, and the anisotropy parameter $\Gamma = 2.5$. The values of the parameters selected here are typical of offshore wind conditions at the FINO1 site for neutral conditions (Syed and Mann, 2023). The 3D turbulent wind field is also generated by the procedure presented in Mann (1998) but the dimensions of the simulated grid are kept small due to the higher computational effort required for generating 3D wind fields. Since the wind fields are assumed statistically independent, they can be added to get the combined fluctuations. In this case, a smaller section of the 2D wind field components is directly added to all the vertical planes of the corresponding 3D turbulence box. We can observe the increased variance in the combined 2D and 3D wind fluctuations. The large-scale coherent structures are still dominant, but now we also observe smaller structures. The one-dimensional two-sided spectra of the 3D turbulence wind field by itself and combined with the





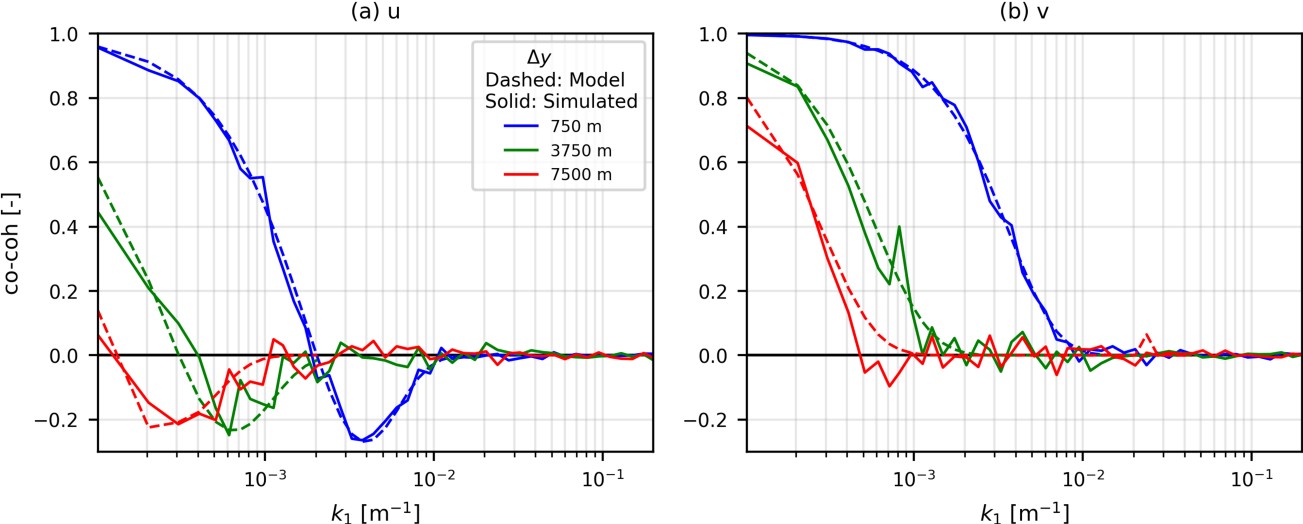

**Figure 4.** Co-coherence of $u$ and $v$ fluctuations at different $\Delta y$ separations for the 2D wind field shown in Fig. 3. The dashed curves show theoretical values, and solid curves show simulated values

low-frequency fluctuations are shown in Fig. 6 (b) and (c), respectively. The resulting spectra are just an addition of individual 2D and 3D wind field spectra over the wavenumber domain.

The $u$ and $v$ co-coherences of simulated combined 2D+3D wind field at different lateral and vertical separations are illustrated in Fig. 7. The co-coherences are plotted for lateral and vertical separations ranging from 150 to 450 m. At lower $k$ values, the low-frequency fluctuations are fully coherent for all vertical $\Delta z$ separations, and we obtain co-coherence values close to 1. This is because the low-frequency fluctuations are assumed to be constant in the vertical direction at any instant. The same can not be said about the lateral $\Delta y$ separations, as we have observed a decrease in the $u$ co-coherence of low-frequency

fluctuations for increasing lateral separations in Fig. 7(a).

## 5 Discussion

### 5.1 The effect of anisotropy parameter on 2D turbulence

As mentioned earlier, $\psi = 45°$ represents isotropic 2D wind fields. Altering the $\psi$ parameter by decreasing or increasing it from $45°$ leads to the elongation of significant coherent structures, extending them longitudinally and laterally, respectively.

The effect of changing the anisotropy parameter $\psi$ can be observed in Fig. 8. Here $u$ and $v$ fluctuations are shown on a 15 km $\times$ 15 km grid for different values of $\psi$. Fig. 8(a) shows the $u$ and $v$ fluctuations for $\psi = 20°$, and we can observe the large-scale coherent structures in the longitudinal direction. These structures exhibit significantly larger values of fluctuations, i.e. $\sigma_u^2 > \sigma_v^2$. Fig. 8(b) illustrates the isotropic case when both $u$ and $v$ fluctuations have similar strength and $\sigma_u^2 = \sigma_v^2$. By

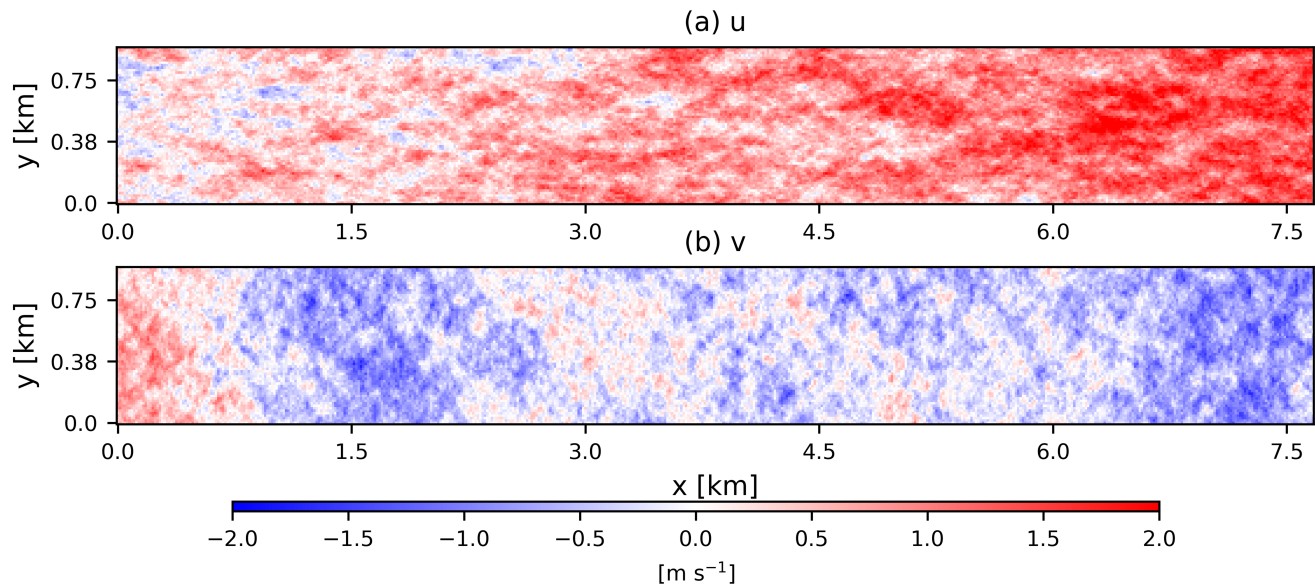

**Figure 5.** Combined 2D+3D fluctuations of longitudinal $u$ and transverse $v$ wind components. The 2D turbulence parameters are the same as in Fig. 3. 3D turbulence parameters are $\alpha\epsilon^{2/3} = 0.01$ m$^{4/3}$s$^{-2}$, $L_{3D} = 50$ m, and $\Gamma = 2.5$

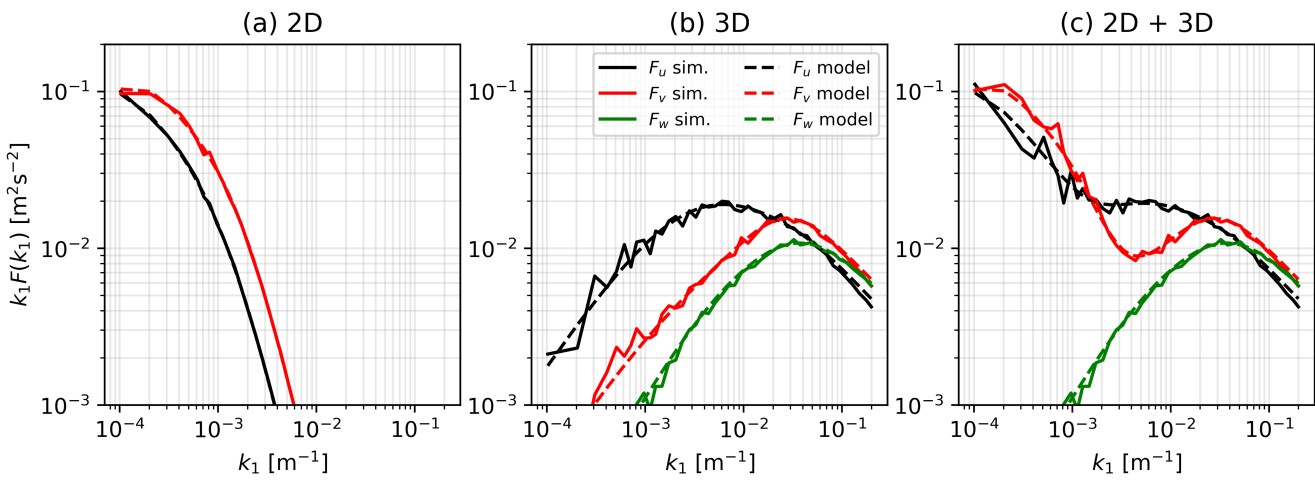

**Figure 6.** Spectra of 2D, 3D, and combined 2D+3D fluctuations of longitudinal $u$ and transverse $v$ wind components. Solid lines present the simulated spectra, and dashed lines reflect the theoretical spectra. The $w$ spectra for 3D turbulence is also shown



**Figure 7.** $u$ and $v$ co-coherences at different $\Delta y$ and $\Delta z$ values for combined 2D+3D fluctuations. The dashed curves show theoretical values, and solid curves show simulated values





increasing the value of $\psi$ to $70°$ (Fig. 8(c)), the large-scale coherent structures in the lateral direction get stretched and we also

observe $\sigma_u^2 < \sigma_v^2$.

## 5.2   Guidelines for simulating 2D wind-fields

Usually $\sigma_{2D}^2$, and $\psi$ are obtainable at a specific site through fitting (9) and (10) to the measured spectra. For $z_i$, some advanced measurements like ground-based remote sensing tools such as a ceilometer can be used, or it can be obtained through reanalysis data sets or simply estimated. To obtain $L_{2D}$ through measurements, we would need a time series spanning from 10 days to 1

180   month (see Fig. 3 in Larsén et al. (2016)). This suggests $L_{2D}$ to be in the order of $10^5$-$10^6$ m. Such extremely low frequencies corresponding to $L_{2D}$ are not interesting for the wind turbine design process. For fitting (9) and (10) to the measured spectra we can assume $L_{2D} \to \infty$. But for wind field simulation purposes, this is not realistic since it would lead to $\sigma_{2D}^2 \to \infty$. For load estimation on wind turbine structures, a 1-hr time series is usually sufficient for estimating the impact of low-frequency fluctuations. Hence, an arbitrarily high value of $L_{2D}$ can be used to simulate low-frequency wind fluctuations. An example of

this is shown in Fig. 9 where a value of $L_{2D} = 150$ km is used to plot the theoretical spectra in (9) and (10) over the $u$ and $v$ spectra measured at FINO1 test site.

An unwanted effect of the simulation method presented here is the periodicity in wind fluctuations, which was also discussed by Mann (1998). The periodicity implies that wind fluctuations at grid points on either side of the box $j$ and $N_2 - j + 1$ for small $j$ are coherent. This behavior is shown in Fig. 10 where co-coherence of $u$ fluctuations is plotted as a function of lateral

distance. It can be observed that both the simulated and model co-coherence values decrease when $y$ approaches $L_2/2$. Due to periodicity, the simulated co-coherence increases for $y > L_2/2$. The solution to this problem is choosing $L_2$ at least twice the characteristic length of the structure under analysis. In the case of wind turbines, $L_2$ should be at least greater than twice the rotor diameter of the wind turbine. A good practice is to simulate the low-frequency fluctuations on a much larger grid than the high-frequency fluctuations. To combine the 2D and 3D turbulence, a smaller section of the 2D wind field, equal in length and

grid points to the 3D turbulence plane, is added to all the vertical levels of the 3D turbulence box.

## 6   Conclusions

A method to generate the low-frequency wind fluctuations is introduced. This method utilizes the spectral tensor presented by Syed and Mann (2023) to generate 2D stochastic wind fields for the longitudinal $u$ and lateral $v$ wind components. The generated wind fields contain large-scale and low-frequency wind fluctuations, also called 2D turbulence. The model employs

four input parameters: (i) the variance characterizing low-frequency fluctuations $\sigma_{2D}^2$, (ii) a length scale corresponding to large-scale flow structures $L_{2D}$, (iii) an anisotropy parameter $\psi$, and (iv) a cutoff or attenuation length $z_i$. The simulation method uses the wind field presented as a discrete Fourier series, where Fourier coefficients are derived from the 2D spectral model. The coefficients are then multiplied by a random Gaussian field. Subsequently, the product's inverse discrete Fourier transform yields a 2D wind field featuring low-frequency, anisotropic wind fluctuations. Issues arising from the discretization,

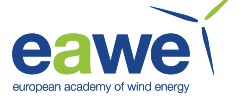
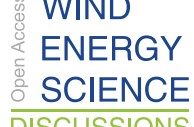


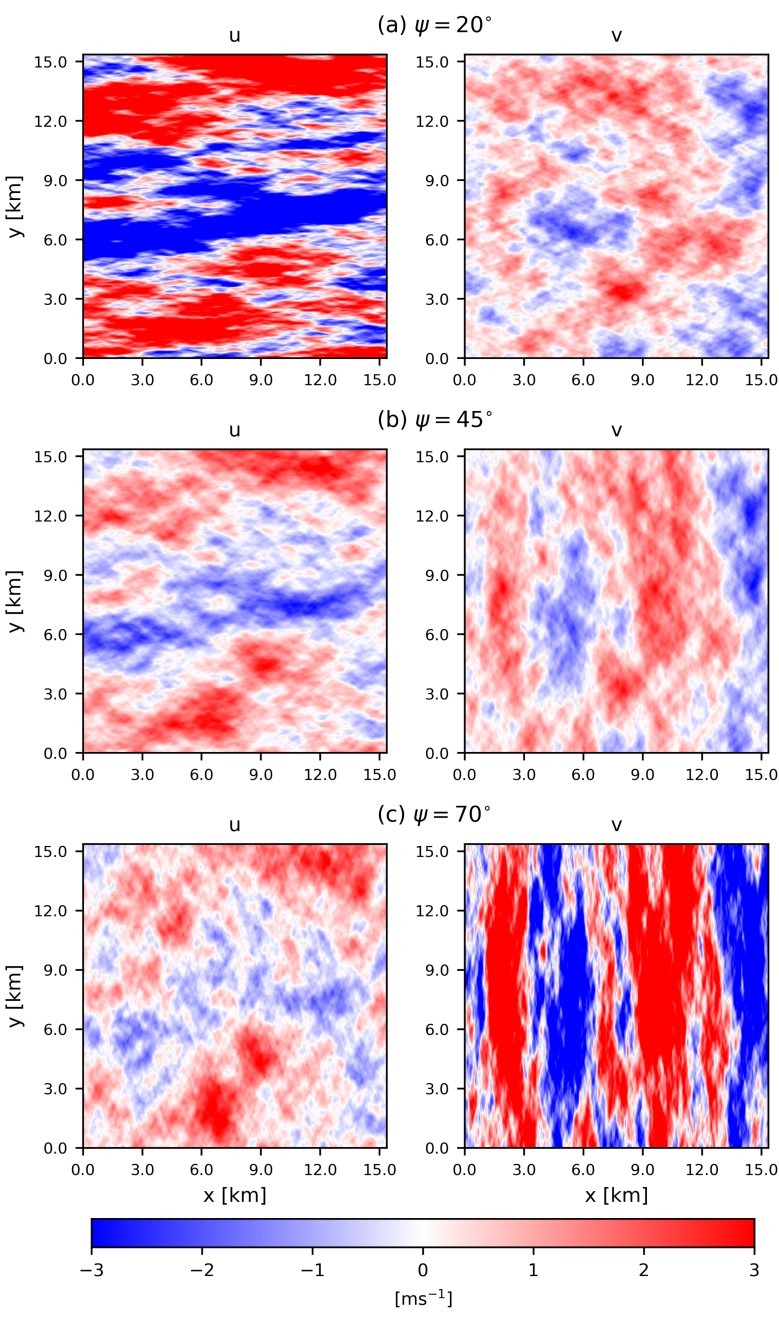

**Figure 8.** Effect of anisotropy parameter $\psi$ on the wind field: (a) $\psi = 20°$, (b) $\psi = 45°$ (isotropic turbulence), and (c) $\psi = 70°$. Here the other input parameters are: $L_{2D} = 5$ km, $\sigma^2 = 1$ m$^2$s$^{-2}$, $z_i = 100$ m



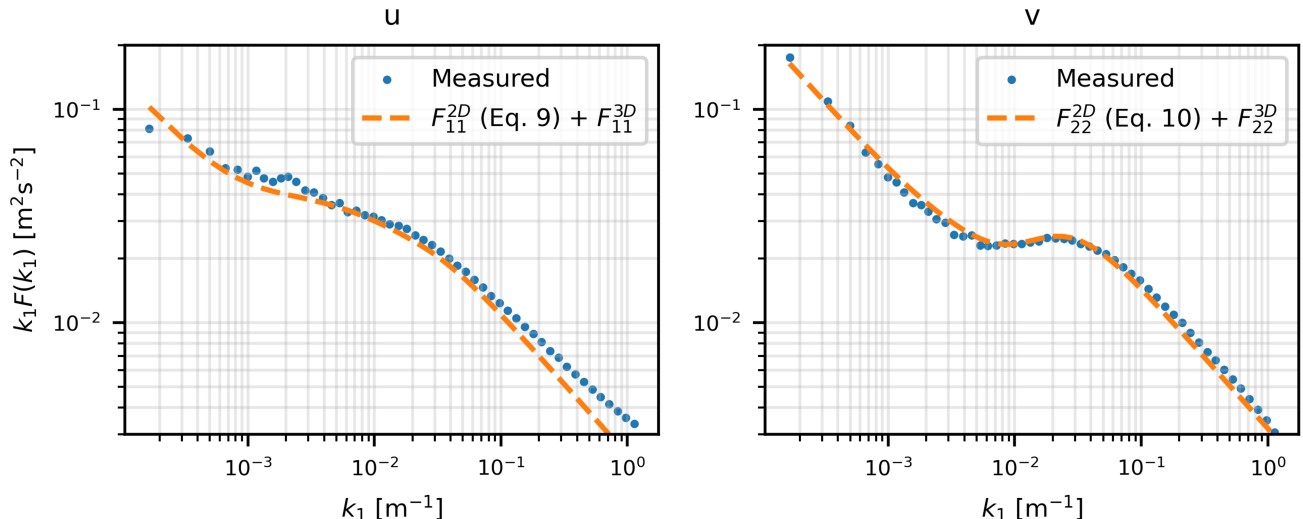

**Figure 9.** An example of theoretical spectra plotted over measured spectra obtained from FINO1 at $z = 81$ m, and $\overline{U} = 10$ ms$^{-1}$. (9) and (10) are plotted with $L_{2D} = 150$ km and $z_i = 100$ m

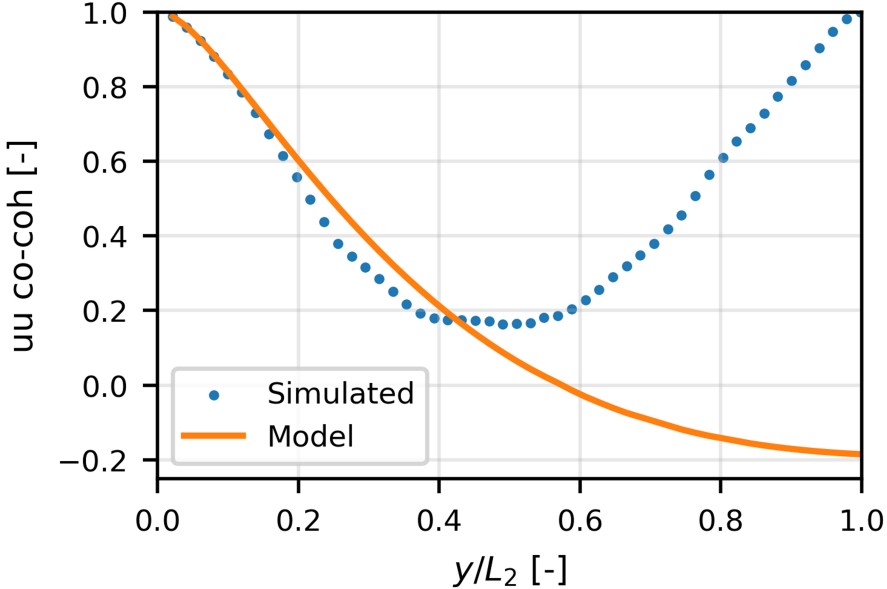

**Figure 10.** An example of periodicity in 2D wind field simulation





such as underestimation of the spectral density at very low wavenumbers and periodicity are also addressed in this study. Some guidelines to simulate the wind fields containing 2D turbulence are also provided in the context of wind energy applications.

The 2D turbulence wind field can be added to a 3D turbulence field to get the spectral representation over a wide frequency range. We combined the 2D turbulence wind field with a 3D turbulence field generated using the Mann uniform shear turbulence model. The spectra and co-coherences from the combined simulated 2D+3D turbulence wind are compared with the theoretical expressions, and an excellent agreement was observed. The 2D turbulence simulation program is open-source and can be accessed via the link in the "Code Availability" section.

*Code availability.* The 2D turbulence simulation program to generate 2D wind fields is available at: GitHub Repository

*Author contributions.* AHS and JM conceptualized and designed the study. AHS wrote the original draft manuscript. JM reviewed and edited the whole manuscript.

*Competing interests.* JM is a member of the editorial board of Wind Energy Science.

*Acknowledgements.* Discussions with Leonardo Alcayaga, DTU Wind Energy are appreciated. We also appreciate the feedback from Arne Rekdal and Marte Godvik of Equinor ASA that helped us improve the code.

*Financial support.* AHS is funded by the European Union Horizon 2020 research and innovation program under grant agreement no. 861291 as part of the Train2Wind Marie Sklodowska-Curie Innovation Training Network (https://www.train2wind.eu/). Funding for JM's work comes from Equinor ASA, and from Atmospheric FLow, Loads and pOwer for Wind energy (FLOW, HORIZON-CL5-2021-D3-03-04, Grant number 101084205), funded by the European Union.



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
