# Peer review of "Simulating low-frequency wind fluctuations"

_Wind Energy Science, 2023_

## Referee Comment (RC2)

**Referee report on "Simulating low-frequency wind fluctuations" by Syed and Mann**

The manuscript provides a sampling algorithm for a recently developed inflow turbulence model [A. H. Syed and J. Mann, A model for low-frequency, anisotropic wind fluctuations and coherences in the marine atmosphere. *Boundary-Layer Meteorology*, 190(1), 1 (2024)] that captures the effects of large-scale anisotropies measured in marine atmospheric boundary layers. The model basically consists of a superposition of a two-dimensional Gaussian random velocity field, which accounts for large-scale anisotropy, and the standard Mann wind field model suggested by the International Electrotechnical Commission.

Overall this is a very well-written and valuable contribution to the literature on inflow turbulence models in the context of wind energy. The open-source code of the sampling algorithm of the anisotropic wind field model should lead to important validation of loads in offshore settings. In general, I support the publication of the manuscript in Wind Energy Science after the following comments are addressed:

i.) Eq. (7) provides a method to determine the degree of anisotropy by measuring the components of the spectral tensor of the two-dimensional velocity field. The wind field model itself, however, consists of a superposition of two- and three-dimensional fields. The authors should add a few words on how the ratio in Eq. (7) can be determined from actual measurement data and refer to their original work.

ii.) The model parameters are listed in line 105. It would be helpful for the reader to list the model parameters determined from the FINO1 and Hywind Scotland measurements in a separate table.

iii.) I would suggest defining the velocity tensor $\phi_{ij}(k_1, k_2)$ in terms of the two-dimensional velocity field $\phi_{ij}(k_1, k_2) = \langle \hat{u}_i(k_1, k_2)\hat{u}_j(-k_1, -k_2)\rangle$, perhaps even based on the velocity field in real space. Please also check all the indices in Eqs. (11-14): I suppose that summation over index $j$ is implied in Eq. (11)? Furthermore, there must be an averaging procedure involved from Eq. (12) to Eq. (13) that should be mentioned.

---

## Author Response (AR1)

**RC 1:**

REVIEW OF Simulating low-frequency wind fluctuations by A.H. Syed and J. Mann

This paper addresses the important problem of generating synthetic turbulent fields. This is an area of significance for wind energy applications and other wind-structure interaction problems. Much effort has been placed on generating 3D fields representing turbulence fine structure including anisotropy and shear effects (RDT), but the arguably even more important aspect of large-scale, slow variations, has been less explored.

The paper provides useful details how to construct the fields and illustrates example results convincingly, culminating in fields where 2D and 3D fields (assumed independent) are successfully superposed (Fig.6). This paper is therefore a welcome addition to the literature. Publication is recommended after the authors take the following comments into account:

(1) The paper often says "low frequency" but means "low wavenumber". It is only if Taylor's frozen flow hypothesis is used and the spatial field is "swept" into a domain does it become frequency. The distinction is important since there are further, more refined models that include both 2 wavenumbers and frequency (see e.g. Wilczek et al (2015), J. Fluid Mech. 769, R1) and references therein). It may be worth stating more explicitly that this work neglects any of those effects and feeds in a "temporally frozen" spatial field. For proper perspective it would be useful if readers are informed that there are more general (but more complicated) "spatiotemporal" models in the literature that have been developed. And avoid saying "low frequency" and replace with "low wavenumber" throughout... On Line 110 you say

"Taylor's frozen turbulence hypothesis is also employed to convert the frequency domain into the wavenumber domain". In the method, it is the reverse, $k_1$ wavenumber domain is being converted into frequency domain. Please correct.

**Authors' Response:**

Thank you for this suggestion. The following comment is now added into the manuscript (See line 99):

> It is important to note that although this model utilizes the wavenumber information to generate a spatial field containing large-scale fluctuations, Taylor's frozen turbulence hypothesis can be used to sweep the spatial field into a frequency domain.

More intricate models, such as those presented by Wilczek et al. (2015) and de Maré and Mann (2016), characterize spatio-temporal turbulence structures as a function of both wavenumber and frequency. However, for the sake of simplicity, the model presented here disregards the temporal variation or distortion of eddies.

(2) The figure 5 shows an "increase" in u' when going from left to right. This is presumably due to the large-scale 2D component. What would be helpful would be to show another panel covering the entire large-scale field (presumably 100 or more of km's) and then showing fig 5a as a "enlarged" portion so we can see that the increase is just a "local" increase in the large-scale part.

**Authors' Response:**

Thank you for pointing this out. In order to avoid displaying a local increase in the large-scale part, we have decided to display the whole 2D+3D wind field with the same dimensions as in Figure 3.

(3) Also in terms of visualization, it would be useful to show the fluctuations also in the z-direction (treated as entirely 3D without any large-scale variation).

**Authors' Response:**

A vertical slice of the combined 2D+3D wind field is now added to the manuscript. See Figure 6.

(4) Is there a way to further improve the method by imposing different large-scale length-scales L for the different velocity components? It is often the case that the L for u component is larger than for the other two components. Here it appears that while the velocity variances are allowed to be anisotropic, the length-scales for these components are still isotropic. Some comments about this issue would be welcome.

**Authors' Response:**

The length scales of the two velocity components are also anisotropic. See the following two figures.

[Figure]

[Figure]

The following text is added in the manuscript. See Paragraph#2 in Section 5.1.

The length scales of the two velocity components can be determined by identifying the maximum of $k_i F_{ii}^{2D}(k_i)$ for $i = 1,2$. Let $L_{i,k_i} \equiv 1/k_{max,i}$, where $k_{max,i}$ denotes the wavenumber at the peak of $k_i F_{ii}^{2D}(k_i)$. These length scales can be computed numerically. At $\psi = 45°$, the ratio $L_{u,k_1}/L_{v,k_2}$ equals 1, indicating that the length scales of $u$ and $v$ are equivalent in the $k_1$ and $k_2$ directions, respectively. When $\psi < 45°$, turbulence structures elongate in the longitudinal direction, resulting in $L_{u,k_1}/L_{v,k_2} > 1$. Conversely, for $\psi > 45°$, the inverse holds true. Moreover, the ratios $L_{u,k_1}/L_{v,k_1}$ and $L_{v,k_2}/L_{u,k_2}$ are independent of the anisotropy parameter. It is noted that these length scale ratios are approximately equal to $\sqrt{3}$, or about 1.73.

Furthermore, here we aimed for the simplicity of the model and just included one length scale parameter. Models with different length scale parameters for different velocity components are often rather complex and difficult to implement. One such example of such a model will be Kristensen et al. (1989) *The Spectral Velocity Tensor for Homogeneous Boundary-Layer Turbulence.*

We thank you for your review of this manuscript. Your comments have greatly improved the quality of this manuscript.

**RC 2:**

**Referee report on "Simulating low-frequency wind fluctuations" by Syed and Mann**

The manuscript provides a sampling algorithm for a recently developed inflow turbulence model [A. H. Syed and J. Mann, A model for low-frequency, anisotropic wind fluctuations and coherences in the marine atmosphere. *Boundary-Layer Meteorology*, 190(1), 1 (2024)] that captures the effects of large-scale anisotropies measured in marine atmospheric boundary layers. The model basically consists of a superposition of a two-dimensional Gaussian random velocity field, which accounts for large-scale anisotropy, and the standard Mann wind field model suggested by the International Electrotechnical Commission.

Overall this is a very well-written and valuable contribution to the literature on inflow turbulence models in the context of wind energy. The open-source code of the sampling algorithm of the anisotropic wind field model should lead to important validation of loads in offshore settings. In general, I support the publication of the manuscript in Wind Energy Science after the following comments are addressed:

  i.) Eq. (7) provides a method to determine the degree of anisotropy by measuring the components of the spectral tensor of the two-dimensional velocity field. The wind field model itself, however, consists of a superposition of two- and three-dimensional fields. The authors should add a few words on how the ratio in Eq. (7) can be determined from actual measurement data and refer to their original work.

**Authors' Response:**

This is now included in the manuscript in Section 2. See the description below Eq. 7 and Eq. 8.

  ii.) The model parameters are listed in line 105. It would be helpful for the reader to list the model parameters determined from the FINO1 and Hywind Scotland measurements in a separate table.

**Authors' Response:**

The model parameters are already discussed in detail in the original paper describing the model i.e. Syed and Mann (2024). We feel that describing them again here is redundant and would take up a lot of useful space.

  iii.) I would suggest defining the velocity tensor $\phi_{ij}(k_1, k_2)$ in terms of the two-dimensional velocity field $\phi_{ij}(k_1, k_2) = \langle \hat{u}_i(k_1, k_2)\hat{u}_j(-k_1, -k_2) \rangle$, perhaps even based on the velocity field in real space. Please also check all the indices in Eqs. (11-14): I suppose that summation over index $j$ is implied in Eq. (11)? Furthermore, there must be an averaging procedure involved from Eq. (12) to Eq. (13) that should be mentioned.

**Authors' Response:**

The two-dimensional velocity spectral tensor is defined as (Syed and Mann, 2024):

$$\phi_{ij}(k_1, k_2) = \int_{-\infty}^{\infty} \int_{-\infty}^{\infty} R_{ij}^{2D}(x_1, x_2)\exp(-\iota k \cdot x)dx_1 dx_2$$
$$= \langle \hat{u}_i^*(k_1, k_2)\hat{u}_j(k_1, k_2)\rangle,$$

Which also utilizes both k1 and k2 domains. Due to the type of data available for model verification and keeping simplicity in mind, we adopted this way.

For Eq. 11 to 14, the summation over repeated indices is assumed. This has now been added to the manuscript. See the description below Eq. 12 (formerly Eq. 11).

Between Eq. 12 and Eq. 13, there is no averaging procedure, rather the coefficients are obtained by calculating the covariance tensor and then applying the convolution theorem to the result. These steps are already mentioned in detail in Mann 1998 and are not repeated here.

We thank you for your review. Your comments and suggestions have improved the quality of this paper.

---

## Author Response (AR2)

Dear Editor and Associate Editor WES,

We have addressed the minor comment from RC2. The reference to corresponding model parameters is now added. See line 106.

Thank you for reading and reviewing this article.